# Nanohydrodynamic Local Compaction and Nanoplasmonic Form-Birefringence Inscription by Ultrashort Laser Pulses in Nanoporous Fused Silica

**DOI:** 10.3390/nano12203613

**Published:** 2022-10-15

**Authors:** Sergey Kudryashov, Alexey Rupasov, Roman Zakoldaev, Mikhail Smaev, Aleksandr Kuchmizhak, Alexander Zolot’ko, Michail Kosobokov, Andrey Akhmatkhanov, Vladimir Shur

**Affiliations:** 1Lebedev Physical Institute, 119991 Moscow, Russia; 2School of Natural Sciences and Mathematics, Ural Federal University, 620000 Ekaterinburg, Russia; 3School of Photonics, ITMO University, 197101 Saint Petersburg, Russia; 4Pacific Quantum Center, Far Eastern Federal University, 690041 Vladivostok, Russia; 5Institute of Automation and Control Processes, Far Eastern Brach of Russian Academy of Sciences, 690041 Vladivostok, Russia

**Keywords:** nanoporous fused silica, ultrashort laser pulses, direct laser inscription, form birefringence, retardance, bulk nanogratings, compaction, interfacial plasmons

## Abstract

The inscription regimes and formation mechanisms of form-birefringent microstructures inside nano-porous fused silica by tightly focused 1030- and 515-nm ultrashort laser pulses of variable energy levels and pulsewidths in the sub-filamentary regime were explored. Energy-dispersion X-ray micro-spectroscopy and 3D scanning confocal Raman micro-spectroscopy revealed the micro-tracks compacted by the multi-shot laser exposure with the nanopores hydrodynamically driven on a microscale to their periphery. Nearly homogeneous polarimetrically acquired subwavelength-scale form-birefringence (refractive index modulation ~10^−3^) was simultaneously produced as birefringent nanogratings inside the microtracks of wavelength-, energy- and pulsewidth-dependent lengths, enabling the scaling of their total retardance for perspective phase-modulation nanophotonic applications. The observed form-birefringence was related to the hierarchical multi-scale structure of the microtracks, envisioned by cross-sectional atomic-force microscopy and numerical modeling.

## 1. Introduction

During the last decade, the ultrashort-pulse laser inscription of (sub)microscale birefringent nanograting arrays in bulk dielectrics has emerged as a new promising key-enabling technology for the non-destructive, re-writable fabrication of innovative low-loss macro-and micro-optical devices–polarizers, phase elements, etc. [1,2,3,4,5]. This technology utilizes the birefringence/polarization effect of the bulk anisotropic nanogratings and polarization/interference effects in their multiple layers with overall retardance approaching wavelength scale [2,5], rather than its common irreversible densification- [6] or void-related refractive-index modulation [7]. These underlying effects provide additional flexibility in terms of the design, internal arrangement, and spatial scales of laser-inscribed functional optical elements and the ongoing progress of this technology [1,2,3,4,5]. Meanwhile, many conjugate physical processes—(non)linear laser focusing [8], nanoplasmonic arrangements [9], and nanoscale material transport mechanisms [10]—in the focal region during material/refractive-index nanograting formation are still poorly understood.

Specifically, compared to the common linear (geometrical) high numerical aperture (NA) focusing of ultrashort laser pulses in bulk dielectrics, non-linear high-NA focusing still involves Kerr-like self-focusing and its usually plasma-based self-consistent clamping in highly elongated filaments [11,12,13]. The filamentary structure of laser-pulse propagation in the near-focal region implies robust and homogeneous photoexcitation, plasmon-supporting plasma density, and the energy deposition of dielectric material [14]. As a result, more productive and reproducible inscription could become possible in the filamentary inscribing of bulk nanograting arrays, but the corresponding laser-focusing regimes were never identified, despite the obvious highly elongated permanent structural modification regions (micro-tracks), made of nanograting arrays [15,16,17,18].

Furthermore, the direct experimental characterization of the key nanograting parameters—refractive index and mass density/chemical concentration modulations—becomes challenging in nanograting arrays owing to their subwavelength nanoscale sub-structure and hierarchical anisotropic arrangement [19], hindering the corresponding sample preparation for their microscopic characterization. In comparison to the low contrast of the smooth microscale appearance of densification on scanning or transmission electron microscopy (SEM, TEM) images [20], the high imaging contrast of bulk nanogratings [15,16,17,18,19] does not necessarily undermine such strong mass-density modulation, as in readily visualized hollow nanovoids [7]. Similarly, the current experimental challenges in the characterization of intrinsic refractive-index modulation in bulk nanogratings impede the understanding of nanoscale material transport mechanisms.

In this study, we employed, for the first time, ultrashort laser pulses of variable wavelengths (515, 1030 nm), widths (0.3–3.8 ps), and energy levels (0.02–0.1 μJ) in a sub-filamentary inscription regime in order to explore the morphological, birefringent, and structural characteristics of the corresponding nanograting arrays in nanoporous fused silica (np-FS) by optical transmission spectroscopy, 3D scanning confocal Raman/photoluminescence microspectroscopy, polarimetric microscopy, and energy-dispersion X-ray microspectroscopy.

## 2. Materials and Methods

In this work, ultrashort-pulse laser inscription in the nanoporous fused silica (np-FS) samples was performed, utilizing 1030 nm and 515 nm laser pulses of variable widths (full-width at half maximum τ ≈ 0.3–3.8 ps varied by the output compressor) and energy levels *E* (varied by a reflective thin-film attenuator, see below). The laser pulses were focused inside the samples at a depth of 50-μm by a 0.25-NA microscope objective (Levenhuk Bioview 630) into the focal spot of the 1/e-radius *R*_1/*e*_ ≈ 1.7 ± 0.1 μm (linear focusing regime), which was continuously moved in the samples linearly translated by a 3D micropositioning stage at a speed of 25 μm/s and a laser repetition rate of 100 kHz (Figure 1a). Three-line structures with a width of 200 μm and a period of 20 μm were inscribed for optical and structural characterization.

Nanoporous fused silica (SiO_2_ content > 95%) samples were prepared by chemical etching, providing an average pore size ~17 nm at an overall volume porosity of 30%. Its UV−near-IR optical transmittance was about 80%, with a cut-off wavelength about 250 nm (Figure 1b).

Our side-view optical microscopy studies of blue recombination plasma emission form the filamentary tracks [21] in the np-FS (Figure 1c, inset) enabled us to identify the critical powers P_cr_ for self-focusing and filamentation in this material, corresponding to the bottom limit pulse energy (P_cr_τ) for the laser filamentation onset of the 1030-nm and 515-nm pulses at different pulsewidths. The entire pulse energy ranges (0.018–0.045 μJ (1030 nm) and 0.05–0.11 μJ (515 nm)) were below the corresponding critical energy P_cr_τ for the utilized pulsewidths of 0.3–3.8 ps (Figure 1c), thus representing the linear (geometrical) focusing regime with the proposed interferential mechanism of nanograting inscription [19,22]. This mechanism implies it is the pre-focal region of interference between the incident laser pulses and their plasma-reflected replica that dictates the nanograting array in-depth length *L*, rather than the corresponding geometrical Rayleigh length [19,22].

The morphological 3D shapes and structural state of fused silica in the 3-line structures were explored by means of 3D scanning confocal Raman/photoluminescence microspectroscopy (Confotec MR350, SOL Instruments). Specifically, the lengths *L* of the modification tracks comprising the buried 3-line structures were measured through 3D scanning with submicron lateral and vertical resolution. The confocal microspectroscopy was performed at the 532 nm laser excitation line from the 1-µm wide focal spot (objective NA = 0.75) in the spectral range of (18 − 20) × 10^3^ cm^−1^ with the 2-cm^−1^ spectral resolution (see Section 3.3).

The microscopic visualization and chemical mapping of silicon (Si) and oxygen (O) distribution within the modification tracks were obtained in their cross-sections, using a field-emission scanning electron microscope VEGA 4 (TESCAN) and its energy-dispersion X-ray micro-spectroscopy module AZTEK (Oxford Instruments). 

The polarimetric acquisition of retardance (Ret) in the 3-line microstructures as a function of ultrashort-pulse laser wavelength, pulsewidth, and energy was carried out by means of a polarimetric microscope Olympus equipped by a probe 633-nm laser, UPlanFl N 10× (NA = 0.3) and 40× (NA = 0.75) microscope objectives, and Abrio software [19,23]. In the polarimetric images, the non-modified np-FS sample demonstrates its intrinsic homogeneous green coloration (Figure 2, Orientation pseudo color mode), corresponding to the background Ret = 42 ± 6 nm, with the slow optical axis at an angle of 60° relative to the horizontal line. This intrinsic birefringence is related to the anisotropic character of surface chemical etching, producing the nanopores during the np-FS fabrication. The polarimetric Abrio characterization provides higher color contrast (Orientation pseudo color) and measurable Ret magnitudes, even for optically invisible 3-line micro-structures inscribed at lower pulse energies or longer pulsewidths. The different background color in Figure 2b,d is due to the fact that different areas of the sample possess different spontaneous birefringence, while the analysis for 515- and 1030-nm inscription wavelengths in terms of Ret magnitude was carried out in different dynamic ranges, which led to a slight shift in color. Note that the difference in the angle of the slow axis orientation in the background area of Figure 2b,d was not significant (<5°).

Finite-difference time-domain (FDTD) calculations were carried out to envision the appearance of the periodic electric-field amplitude modulations upon the reflection of the linearly-polarized plane wave from the plasma, as well as the transverse excitation of the surface plasmons. The photo-excited low-loss plasma located inside the bulk silica (the bare dielectric function value ε_SiO2_ = 2.25) was considered to have the complex dielectric function ε*_515_ = −2.25 + i0.3, in order to fulfill the interfacial plasmon resonance condition. Simulations were performed using the commercial FDTD solver at a mesh size of 1 × 1 × 1 nm^3^ and perfectly matched layers as boundary conditions, limiting the simulated volume.

## 3. Results

### 3.1. Length of Microtracks

The measured microtracks exhibit their lengths *L*, increasing beyond the corresponding Rayleigh length of the focusing 0.25-NA micro-objective (1030 nm–≈12 μm, 515 nm–≈7 μm) versus the increasing pulse energy in the threshold-like manner (Figure 3a,b), in good agreement with the nanoplasmonic mechanism of form-birefringence inscription in the form of bulk nanogratings [19,22]. The corresponding thresholds are pulsewidth- and wavelength-dependent quantities (Figure 3a,b), while the observed length magnitudes are quite comparable for both the 1030-nm and 515-nm wavelengths. Still, the shorter laser pulses produced longer tracks, as compared to the longer ones, at the same pulse energies.

In terms of the laser pulsewidths τ (Figure 3c,d), the observed microtracks are longer for the shorter pulses and higher pulse energies, though without the specific maximum at some intermediate pulsewidth previously observed in fluorite [19].

### 3.2. Birefringent Characteristics of Microtracks

Compared to the background birefringence of the np-FS sample, the laser-modified regions (microtracks) appear very distinctly both in optical and polarimetric microscopic images (Figure 2, dark and yellow pseudo-color three-line structures, respectively). Both for the 1030-nm and 515-nm wavelength with the different laser polarizations in Figure 2, the “slow” optical axis in the inscribed three-line structures is always oriented at the angle of 30° regarding the laser polarization.

The measured *Ret* magnitudes demonstrate pronounced pulsewidth and wavelength effects (Figure 4a,b), usually with much lower values for the longer laser pulses, while 515-nm laser pulses very unexpectedly produced two-fold lower retardance even for few times higher pulse energies. Although the apparent threshold for birefringence inscription in Figure 4a,b is commonly lower for shorter pulses (0.3 ps vs. 1.8 ps and 3.8 ps), against expectations [24,25], it is anomalously much higher for the 515-nm pulses (0.05–0.09 μJ (Figure 4b)), compared to the 1030-nm pulses (0.01–0.03 μJ (Figure 4a)), despite the stronger focus of the visible-range pulses. These unusual trends are discussed in Section 4 in terms of optical losses and related nanoplasmonics in the np-FS material.

In contrast, if the length *L* of the birefringent microtracks (Figure 3) is explicitly taken into account, the derivation for the first time average refractive index change, Δn (633 nm) = *Ret*/*L*, indicates very different dependence on laser pulsewidth τ (Figure 4c,d). Here, the maximal value of Δn_max_ ≈ (4 ± 2) × 10^−3^ at 1030 nm (Figure 4c) and Δn_max_ ≈ (1.5 ± 0.5) × 10^−3^ at 515 nm (Figure 4d) appear pulsewidth-independent but strongly wavelength-dependent, as expected according to the nanoplasmonic mechanism of the form-birefringence inscription in the form of bulk nanogratings [19,22]. Moreover, the maximal values are achieved in the threshold-like manner versus E (Figure 4c,d), as expected within the latter mechanism; they will be discussed below. Finally, such a refractive index change (~10^−3^) in the laser-modified np-FS could be related, in fused silica, to the increased mass density (~1%), e.g., as a result of laser-induced nanoscale densification [26].

### 3.3. Structural Modification inside Microtracks

The multi-shot microscale exposure of the np-FS sample by 1030-nm, 0.3-ps laser pulses induces the distinct compaction of the nanoporous material within the 3-line structures and its rarefaction around them, as revealed by SEM and EDX (Figure 5a–c). The inscribed three-line segment exhibits the pronounced SEM contrast (Figure 5a) and the increased EDX intensity of O- and Si-components (Figure 5b,c)—by 6 rel. % and 2 rel. % (Figure 5d), respectively, relative to the surrounding non-modified material with its original 30% porosity. Such compaction in the segment is naturally accompanied by its peripheral depleted regarding these chemical elements. Overall, the prolonged laser exposure apparently drives the nanoscale hydrodynamic phenomena of partial nanopore collapse in the laser-modified region and lateral material flow into it.

These findings are strongly supported by our Raman microspectroscopic studies (Figure 5e), indicating the significant transformation of the Raman spectrum of the laser-modified np-FS toward monolithic FS. Specifically, even though the np-FS spectra demonstrate more than five-fold lower intensities of D1 (490 cm^−1^) and D2 (603 cm^−1^) bands, representing the three-membered and four-membered defect breathing modes in fused silica [27], their intensities appear almost two-fold higher for the laser-modified np-FS, as compared to the non-modified one. The higher intensity of the 603-cm^−1^ band in the laser-modified np-FS could be related to the increased mass density (~0.1–1%) and refractive index (~10^−3^) in fused silica due to its laser-induced densification [26]. Therefore, in our case of the np-FS samples, both laser-induced partial local compaction and the related refractive-index change could be produced in the chosen inscription regimes and probed by time-resolved non-invasive temperature- and density/pressure-sensitive diagnostics [28,29,30].

## 4. Discussion

To date, the main challenge in the modeling of birefringent nanograting inscription in bulk dielectrics is the tailoring planar nanoplasma configurations, which could support excitation by the linearly polarized laser pulses and the following propagation of interfacial plasmons or plasmon-polaritons [19]. In this study, our experimental findings imply that in the linear focusing regime, bulk nanostructuring in fused silica could proceed via the formation of reflective near-critical focal plasma, its significant reflection/backscattering of the incident ultrashort laser pulses, and the interferential formation of a 1D longitudinal standing electromagnetic wave in the pre-focal region [19,22]. According to our full-wave FDTD analysis (see Materials and Methods section), the low-loss plasma, the spherical-shape inclusion with its dielectric function ε*_515_ = −2.25 + i0.3 at the 515-nm wavelength, provides pronounced radially longitudinal standing waves in the pre-focal region with a characteristic period of λ/2*n* (Figure 6a), where *n* is the refractive index of the dielectric.

The dynamic 1D longitudinal standing electromagnetic wave produces a spatial series of dynamic local periodic 2D-nanoplasmas (longitudinal array of plasma 2D-nanosheets with the period Λ ≈ λ/2*n* and thickness δ < Λ; Figure 6b) via the non-linear photoionization of np-FS. These nanoplasmas could be imprinted in the material as periodic tracks [9] or voids, as seen in the AFM image provided in Figure 6c. Moreover, these nanoplasmas during the same ultrashort laser pulse could support laser excitation of interfacial 1D-plasmons (wavenumber κ), propagating in the 2D-nanosheets along the laser polarization [19,22]. Then, the counterpropagating interfacial plasmons with the opposite wave-vectors can dynamically interfere with each other, inducing in each plasma nanosheet a dynamic 1D standing electromagnetic wave with a deeply subwavelength period ν~1/(2κ) and corresponding dynamic transverse plasma nanogratings. The appearance of such nanogratings was justified by our FDTD analysis, showing the distribution of the transverse (E_y_) component of the electric field near the nanosheet surface (bottom panel, Figure 6b). As a result, a permanent birefringent 1D material nanograting can follow such dynamic transverse 1D plasma nanograting in each 2D plasma nanosheet of its longitudinal array, resulting in the final hierarchical, multi-scale form-birefringent microstructures.

Within this nanoplasmonic mechanism [19,22], one could explain the experimentally observed extraordinary higher inscription threshold energies and lower average refractive-index change Δn_633_ for the 515-nm ultrashort laser pulses, which usually more strongly and non-linearly interact with dielectric materials and, specifically, fused silica, as compared to 1030-nm pulses [24,25]. Particularly, such higher non-linear optical losses could considerably dampen the interfacial plasmons driven by the 515-nm laser pulses, making them near- or weakly sub-wavelength [31]. Hitherto, the corresponding 515 nm laser-induced material and refractive-index modulations will be weaker at longer (near-wavelength) scales, as compared to the corresponding 1030-nm laser-induced modulations, in good agreement with our experimental findings.

## 5. Conclusions

In this study, the inscription of form-birefringent microstructures inside nano-porous fused silica by tightly focused 1030- and 515-nm ultrashort laser pulses of varying energy levels and pulsewidths in the sub-filamentary regime resulted in extraordinary higher inscription threshold energies and lower average refractive-index modulations (~10^−3^) for the visible-range pulses. This anomaly was interpreted, considering the hierarchical multi-scale nano/microstructure of the laser-modified birefringent regions and their nanoplasmonic nature, which is very sensitive to much stronger non-linear optical losses in the material at the 515-nm laser wavelength. Structurally sensitive energy-dispersion X-ray micro-spectroscopy and 3D scanning confocal Raman micro-spectroscopy revealed the local considerable compaction of the birefringent regions via partial nanopore collapse and hydrodynamic melt flows. The laser wavelength-, energy- and pulsewidth-dependent lengths of the birefringent microtracks enable the precise tuning of their total retardance for perspective phase-modulation nanophotonic applications.

## Figures and Tables

**Figure 1 nanomaterials-12-03613-f001:**
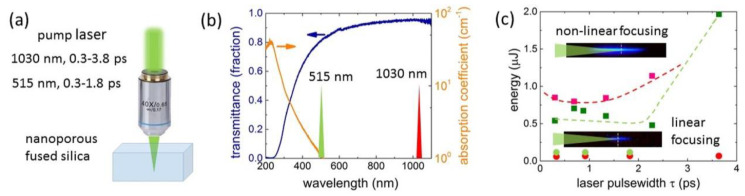
(**a**) Schematic of the laser setup. (**b**) UV-near-IR transmittance and absorption coefficient spectra of the np-FS sample. (**c**) Critical energy values P_cr_τ for non-linear focusing (Kerr self-focusing) in np-FS versus τ at 1030 nm (red squares and dashed curve) and 515 nm (green squares and dashed curve) wavelengths in comparison to the maximal pulse energy values used in the study (bottom red and green circles). Inset: side-view visualization of linear and non-linear focusing via blue recombination plasma emission.

**Figure 2 nanomaterials-12-03613-f002:**
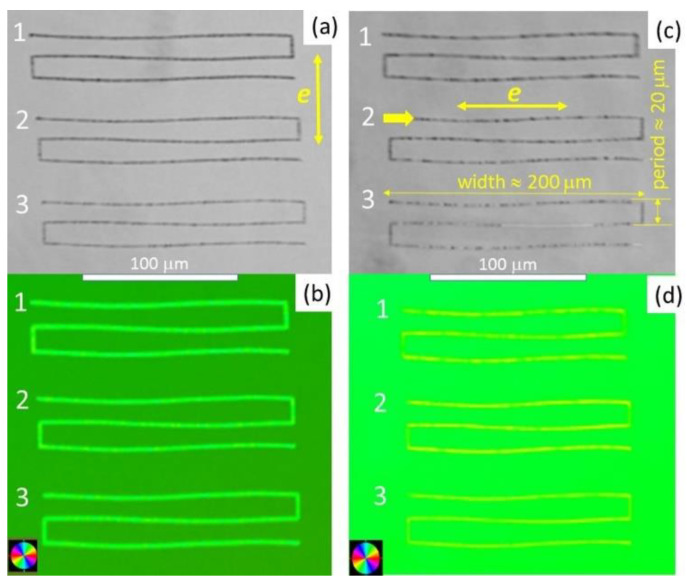
(top) Optical images of typical 3-line 200-μm wide structures (period-20 μm) inscribed in nanoporous fused silica at a depth of 50-micron, at 1030-nm (**a**) and 515-nm (**c**) wavelengths; 0.3-ps laser pulsewidths; and 0.045 (1), 0.038 (2), 0.033 (3) and 0.11 (1), 0.10 (2), and 0.09 (3) μJ energy levels, respectively. The large yellow arrows indicate the laser polarization direction. (bottom) Their corresponding polarimetric (Abrio) pseudocolor images (**b**,**d**), where the color of the lines shows their “slow” axis direction according to the color maps.

**Figure 3 nanomaterials-12-03613-f003:**
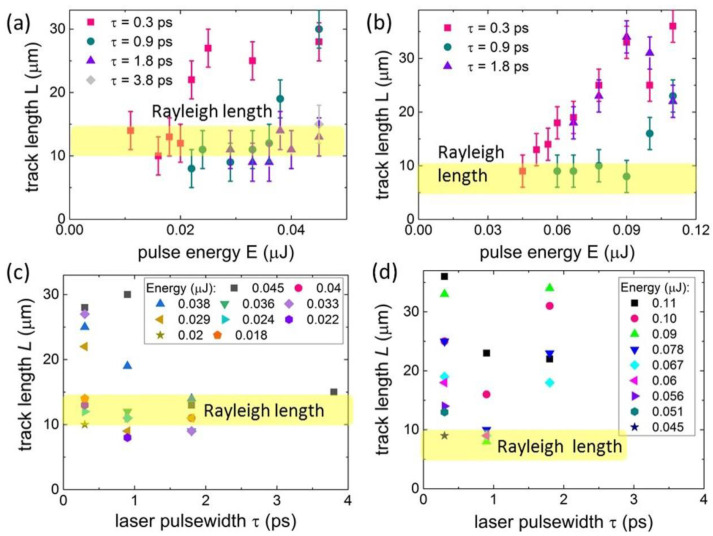
Microtrack length *L* as a function of pulse energy *E* at different pulsewidths τ (**a**,**b**) and of pulsewidth τ at different pulse energies *E* (**c**,**d**) at 1030-nm (**a**,**c**) and 515-nm (**b**,**d**) wavelengths (for error bars see (**a**,**b**). The yellow band highlights the Rayleigh length at these laser wavelengths for the 0.25-NA micro-objective in the linear focusing regime.

**Figure 4 nanomaterials-12-03613-f004:**
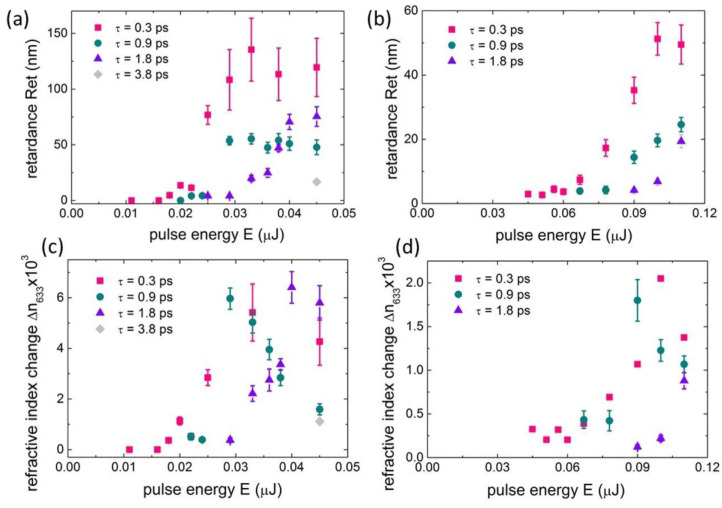
Background-corrected retardance *Ret* (**a**,**b**) and average refractive-index change Δn_633_ (**c**,**d**) versus pulse energy at different laser pulsewidths at 1030-nm (**a**,**c**) and 515-nm (**b**,**d**) wavelengths.

**Figure 5 nanomaterials-12-03613-f005:**
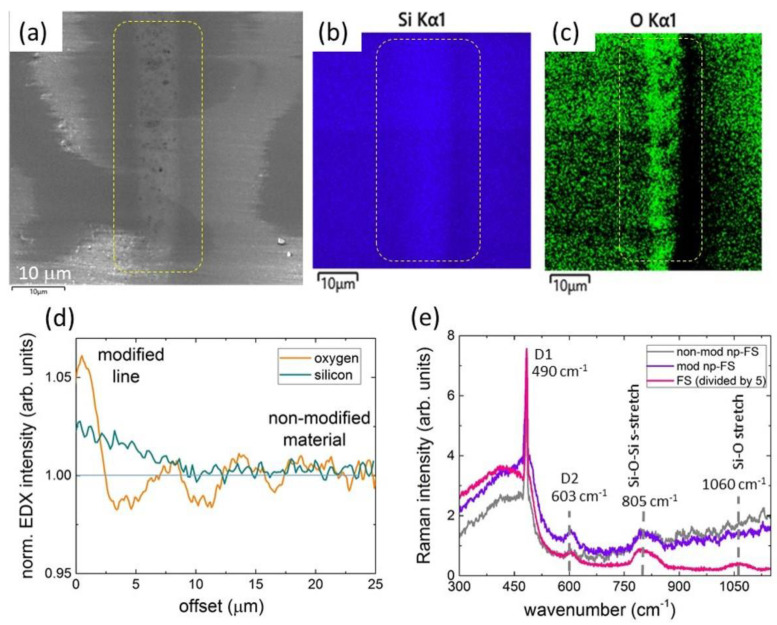
(**a**) Cross-sectional top-view SEM image of three-line segment in np-FS (1030 nm, 0.3 ps. 0.03 μJ). (**b**,**c**) EDX maps of oxygen and silicon, indicating compaction in the segment. (**d**,**e**) Raman spectra of the segment and background in np-FS in comparison to reference spectrum of bulk FS (spectral assignment after [27]).

**Figure 6 nanomaterials-12-03613-f006:**
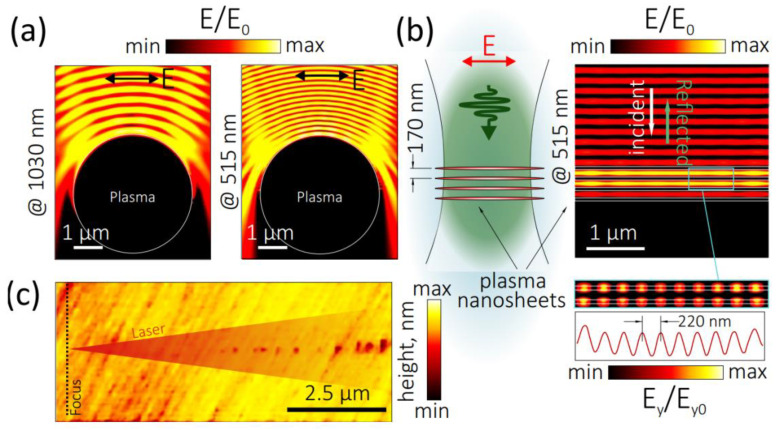
(**a**,**b**) Normalized electrical field amplitude (E/E_0_) calculated in front of the focal plasma (ε*_515_ = −2.25 + 0.3i) in fused silica upon its irradiation by a x-polarized plane wave at 1030-nm (left panel) and 515-nm (right panel) wavelengths. The incident and reflected waves produce radially longitudinal standing wave distribution with a characteristic period of λ/2*n* in front of the plasma. (**b**) Schematically illustrated periodically arranged pre-focal 2D-plasma nanosheets (left panel) as well as calculated (E/E_0_) distribution in their vicinity upon their exposure at 515 nm (top right panel). Bottom panel shows normalized y-component of the electrical field amplitude (E_y_/E_0_) in the vicinity of nanosheets surface with a characteristic modulation period of 220 nm. (**c**) AFM cross-sectional height image of the pre-focal longitudinal array of relief nanobits produced by the tightly focused 1030-nm 0.3-ps laser radiation.

## Data Availability

The data supporting the reported results can be obtained from the authors.

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
