# Peer review of "Nanohydrodynamic Local Compaction and Nanoplasmonic Form-Birefringence Inscription by Ultrashort Laser Pulses in Nanoporous Fused Silica"

_nanomaterials, 2022, doi:10.3390/nano12203613_

Round 1

Reviewer 1 Report

Manuscript reference number: nanomaterials-1949643

Title: Nanohydrodynamic local compaction and nanoplasmonic form-birefringence inscription by ultrashort laser pulses in nanoporous fused silica by Sergey Kudryashov et al

The paper is rather interesting, and well organized. It could be published on this journal but after major revisions according to the list of the following critical points: 

a) The abstract should be revised. At the end of the abstract there is a description of the hierarchical multi-scale structure of the microtracks that could be shifted in the introduction. On the contrary at the beginning of the abstract it is missing a brief introduction to explain clearly what is the aim of the article.

b) Figure 1 is unclear: the authors should increase and make uniform the size and the colors of the labels and titles of the axes

c) Figure 3: the authors should increase and make uniform the size and the colors of the labels and titles of the axis. Moreover in figure 3d  the area highlighted in yellow covers the legenda.

d) The thermophysical properties of the materials are missing in the paper. We believe that it could be important at least to cite some techniques to test these properties. Relevant papers to be cited to enrich the bibliography are for example:

1. O.B.Wright et al., J. Appl. Phys. 91, 5002-5009 (2002).

2. M. Tomoda, et al.  Appl. Phys. Lett.  91, 071911  (2007).

3. F.R. Lamastra, et al. International Journal of Thermophysics, 39, 110, (2018).

Author Response

The paper is rather interesting, and well organized. It could be published on this journal but after major revisions according to the list of the following critical points: 

a) The abstract should be revised. At the end of the abstract there is a description of the hierarchical multi-scale structure of the microtracks that could be shifted in the introduction. On the contrary at the beginning of the abstract it is missing a brief introduction to explain clearly what is the aim of the article.

We are thankful to the Reviewer for this suggestion. Abstract was revised. 

b) Figure 1 is unclear: the authors should increase and make uniform the size and the colors of the labels and titles of the axes

 Changed

c) Figure 3: the authors should increase and make uniform the size and the colors of the labels and titles of the axis. Moreover in figure 3d the area highlighted in yellow covers the legenda.

 Changed

d) The thermophysical properties of the materials are missing in the paper. We believe that it could be important at least to cite some techniques to test these properties. Relevant papers to be cited to enrich the bibliography are for example:

  1. O.B.Wright et al., J. Appl. Phys. 91, 5002-5009 (2002).
  2. M. Tomoda, et al.  Appl. Phys. Lett.  91, 071911  (2007).
  3. F.R. Lamastra, et al. International Journal of Thermophysics, 39, 110, (2018).

Introduced in the relevant discussion of thermal densification.

Reviewer 2 Report

The submitted paper discusses an interesting phenomenon: the inscription of nanograting arrays in nanoporous fused silica with laser pulses of different parameters. The authors present experimental results and model calculations too. The topic is interesting, and the implementation of the study is adequate. However, I have some questions and suggestions for the authors:

1.      Please find uploaded the paper with indications of suggested language corrections. Simplified or better-structured sentences at some points could improve the readability of the text. Since the reviewer is not a native speaker either, it is also suggested to have the text checked by a language expert.

2.      Figure 1 a): consider indicating the scanning direction, the length of the 3-line structure and L, the track length on the figure. It is a bit perplexing that — although they are perpendicular to each other, as I understand — both are called length. L could be called the track depth too.

3.      Figure 2: Why does the unaffected np-FS appear in different colours in Fig. 2. (b) and (d)?

4.      Figure 4: What is the reason for the sudden increase of the standard deviation of the retardance caused by 3 ps pulses above 0.03 mJ?

5.      Figure 6: (b) suggests visually that the plasma nanosheets are behind the focal plane, while they are supposed to be in front of that, looking from the same direction as k of the incident pulse. How was the cross-section shown in Fig. 6 (c) cut? The text does not say anything about this part of the experiment, although it is not straightforward that the cross-section can preserve the array of relief nanobits after processing. Are the tilted lines related to the plasma nanosheets or the cross-section preparation? If the former is true, why are they tilted with respect to the focal plane?

6.      Line 231: Please add a reference or reasoning of using the given value of the complex dielectric constant of the spherical plasma inclusion at 515 nm. 

Author Response

The submitted paper discusses an interesting phenomenon: the inscription of nanograting  arrays in nanoporous fused silica with laser pulses of different parameters. The authors present experimental results and model calculations too. The topic is interesting, and the implementation of the study is adequate. However, I have some questions and suggestions for the authors:

  1. Please find uploaded the paper with indications of suggested language corrections. Simplified or better-structured sentences at some points could improve the readability of the text. Since the reviewer is not a native speaker either, it is also suggested to have the text checked by a language expert.

We have followed the corrections in peer-review-22834815.v1.pdf to achieve smoother reading.

  1. Figure 1 a): consider indicating the scanning direction, the length of the 3-line structure and L, the track length on the figure. It is a bit perplexing that — although they are perpendicular to each other, as I understand — both are called length. L could be called the track depth too.

Additional info was added in Fig.2: track width – 200 microns, period – 20 microns.

  1. Figure 2: Why does the unaffected np-FS appear in different colours in Fig. 2. (b) and (d)?

The different background color in Figures 2b,d is due to the fact that different areas of the sample possess different initial spontaneous birefringence, while the analysis for 515- and 1030-nm inscription wavelengths in terms of Ret magnitude was carried out in different dynamic ranges, which led to a slight shift in color. Note that the difference in the angle of the slow axis orientation in the background area of Figures 2b,d was not significant (< 5°).

  1. Figure 4: What is the reason for the sudden increase of the standard deviation of the retardance caused by 3 ps pulses above 0.03 mJ?

The microtracks inscribed at the pulse duration of 3 ps and energies above 0.03 mJ turned out to be more inhomogeneous (“rough”) than for other durations or energies. This results in the larger standard deviation for these experimental conditions.

  1. Figure 6: (b) suggests visually that the plasma nanosheets are behind the focal plane, while they are supposed to be in front of that, looking from the same direction as kof the incident pulse.

Not exactly, Fig.6b schematically indicates the plane plasma nanosheets, representing the inclined standing waves in Fig.6a (where the inclination depends on focusing NA).

How was the cross-section shown in Fig. 6 (c) cut? The text does not say anything about this part of the experiment, although it is not straightforward that the cross-section can preserve the array of relief nanobits after processing.

It was diamond-saw cut through the segments and polished then prior AFM characterization.

 Are the tilted lines related to the plasma nanosheets or the cross-section preparation? If the former is true, why are they tilted with respect to the focal plane?

Yes, the titled scratches in Fig.6c appear during cutting/polishing.

  1. Line 231: Please add a reference or reasoning of using the given value of the complex dielectric constant of the spherical plasma inclusion at 515 nm. 

Added to define the interfacial plasmon resonance condition in the plasma sheets -eSiO2=e*515 (where eSiO2=2.25, e*515=-2.25+i0.3 [19,31]).

Round 2

Reviewer 1 Report

The manuscript has been revised according to the reviewer suggestions. We guess it is now much more improved and might be published in the present form